# β-Cryptoxanthin Attenuates Cigarette-Smoke-Induced Lung Lesions in the Absence of Carotenoid Cleavage Enzymes (BCO1/BCO2) in Mice

**DOI:** 10.3390/molecules28031383

**Published:** 2023-02-01

**Authors:** Rachel A. Chiaverelli, Kang-Quan Hu, Chun Liu, Ji Ye Lim, Michael S. Daniels, Hui Xia, Jonathan Mein, Johannes von Lintig, Xiang-Dong Wang

**Affiliations:** 1Nutrition and Cancer Biology Laboratory, Jean Mayer USDA Human Nutrition Research Center on Aging, Tufts University, Boston, MA 02111, USA; 2Biochemical and Molecular Nutrition Program, Friedman School of Nutrition Science and Policy, Tufts University, Boston, MA 02111, USA; 3Department of Pharmacology, School of Medicine, Case Western Reserve University, Cleveland, OH 44106, USA

**Keywords:** β-cryptoxanthin, vitamin A, lung inflammation, cigarette smoke exposure, carotenoid cleavage enzymes

## Abstract

High dietary intake of β-cryptoxanthin (BCX, an oxygenated provitamin A carotenoid) is associated with a lower risk of lung disease in smokers. BCX can be cleaved by β-carotene-15,15′-oxygenase (BCO1) and β-carotene-9′,10′-oxygenase (BCO2) to produce retinol and apo-10′-carotenoids. We investigated whether BCX has protective effects against cigarette smoke (CS)-induced lung injury, dependent or independent of BCO1/BCO2 and their metabolites. Both BCO1^−/−^/BCO2^−/−^ double knockout mice (DKO) and wild type (WT) littermates were supplemented with BCX 14 days and then exposed to CS for an additional 14 days. CS exposure significantly induced macrophage and neutrophil infiltration in the lung tissues of mice, regardless of genotypes, compared to the non-exposed littermates. BCX treatment significantly inhibited CS-induced inflammatory cell infiltration, hyperplasia in the bronchial epithelium, and enlarged alveolar airspaces in both WT and DKO mice, regardless of sex. The protective effects of BCX were associated with lower expression of IL-6, TNF-α, and matrix metalloproteinases-2 and -9. BCX treatment led to a significant increase in hepatic BCX levels in DKO mice, but not in WT mice, which had significant increase in hepatic retinol concentration. No apo-10′-carotenoids were detected in any of the groups. In vitro BCX, at comparable doses of 3-OH-β-apo-10′-carotenal, was effective at inhibiting the lipopolysaccharide-induced inflammatory response in a human bronchial epithelial cell line. These data indicate that BCX can serve as an effective protective agent against CS-induced lung lesions in the absence of carotenoid cleavage enzymes.

## 1. Introduction

Cigarette smoke (CS) is one of the leading preventable risk factors for inflammatory lung diseases [1], including chronic obstructive pulmonary disease (COPD) [2] and lung cancer [3]. Although the global prevalence of smoking has decreased from 26% to 19% among adults, the number of current smokers has increased along with population growth [4]. About 30% of smoking-related lung cancer patients continue to smoke after diagnosis. Passive smokers (exposed to environmental cigarette smoke—smoker’s spouse and children) are also at an increased risk of lung disease. All of this emphasizes the importance of developing an effective dietary preventive agent against CS-related chronic disease risk.

Epidemiological studies have inversely linked high consumption of fruits and vegetables rich in carotenoids and serum levels of carotenoids with decreased risk of lung cancer and COPD in smokers [5,6]. Despite numerous in vitro and animal studies that demonstrate the mechanisms of action of carotenoids, the metabolism and molecular biologic properties of many carotenoids remain to be determined [7]. β-cryptoxanthin (BCX), an oxygenated provitamin A carotenoid found in red sweet peppers, butternut squash, and tangerines and one of the six most abundant carotenoids in human plasma and tissues, is a promising agent against lung cancer [7,8]. In contrast to β-carotene trials among smokers [7], which proved ineffective and harmful at high doses, in the Alpha-Tocopherol, Beta-Carotene Cancer Prevention (ATBC) trial, the Carotene and Retinol Efficacy Trial (CARET), data from National Health and Nutrition Examination Survey III (NHANES III) [9], and other epidemiological studies [10,11], including seven large well-implemented cohorts [12], high serum levels of BCX were associated with a lower risk of lung cancer, independent of intakes of other nutrients, such as vitamin C, folate, and other carotenoids, including β-carotene, α-carotene, lutein, lycopene, and zeaxanthin [9,10,11,12]. Indeed, animal experimental evidence showed that BCX supplementation was effective at reducing the lung tumorigenesis in a CS-exposed ferret model and lung cancer AJ mouse model [13,14]. These data support the notion that BCX has a unique beneficial effect against CS-related lung lesions compared to β-carotene. However, these studies raised the question of whether the effects of BCX on various cellular functions and signaling pathways are results of the actions of BCX or its enzymatic cleavage metabolites.

BCX, as a β-carotene, can be cleaved by β-carotene-15,15′-oxygenase (BCO1) at the 15,15′ double bond, which is critical for vitamin A (apo-15-carotenoids) production and homeostasis [15,16,17]. The second enzyme, β-carotene-9′,10′-oxygenase (BCO2), cleaves BCX at the 9′,10′ double bond to generate both apo-10′-carotenoids and 3-hydroxyl (3-OH) apo-10′-carotenoids [18,19,20], and apo-10′-carotenoids can be further cleaved by BCO1 to generate vitamin A [21]. In addition to the well-known biological actions of vitamin A, apo-carotenoids produced from the BCO2 cleavage of carotenoids also possess biological activities similar to their parent carotenoids or have entirely different functions, which can have either beneficial or detrimental effects to human health [7,22]. Genetic variants of the BCO1/BCO2 genes have been associated with alterations in the status of human and animal carotenoid levels and function of provitamin A carotenoids [23]. Single nucleotide polymorphisms (SNPs) in the human BCO1 gene are common and associated with reduced catalytic activity in the conversion of β-carotene to vitamin A [24]. An SNP in BCO2 was strongly related to age-related eye disease [25], to alterations in pro-inflammatory cytokine IL-18 expression, and to fasting high-density lipoprotein cholesterol levels [26,27,28]. Several animal studies have found differential lipid, cholesterol, and oxidative-stress-related physiological abnormalities in individual BCO1 and BCO2 knock out (KO) mouse models [29,30,31]. These studies raised the question of whether the BCO1/BCO2 genotype could play a role in contributing to the adverse effects of β-carotene supplementation in smokers. Whether the presence or absence of BCO1 and BCO2 differentially affects CS-exposure-induced lung lesion severity has not been investigated.

Since previous studies have demonstrated that ablating BCO1 resulted in over-expression of BCO2 [29,30], we used the BCO1/BCO2 double knock out (DKO) mouse model and demonstrated that BCO1/BCO2 DKO mice developed hepatic steatosis and had significantly higher levels of hepatic and plasma triglycerides and total cholesterol compared to wild type (WT) mice with fully functioning BCO1 and BCO2 [32]. We have shown that BCX supplementation significantly suppressed hepatic tumorigenesis without changing hepatic vitamin A levels in BCO1/BCO2 DKO mice [33]. However, the effect of dietary BCX feeding on CS-exposure-induced lung injury in both BCO1/BCO2 DKO and WT littermates has not been investigated. Although vitamin A (retinol, retinal, and retinoic acid) is well known to serve critical functions in lung disease prevention, the potential biological function of 3-OH-β-apo-10′-carotenal (3OH-BA10C), a specific BCO2 cleavage product from BCX, has not been investigated. 

In the current study, we aimed to understand whether carotenoid cleavage enzymes (BCO1/BCO2 genotype) could influence the effects of BCX supplementation in smokers and examined the effects of dietary BCX feeding on short-term CS-exposure-induced lung injury in BCO1/BCO2 DKO and WT littermates. To further understand whether the effect of BCX is a result of the actions of BCX or its enzymatic cleavage metabolites, we also investigated the effects of BCX and its biologically active BCO2 metabolite, 3OH-BA10C, on lung inflammation in human bronchial epithelial cells (BEAS-2B) in vitro.

## 2. Results

### 2.1. Body Weight and Urinary Cotinine Levels

The final body weight remained unchanged for all groups throughout the experiment (Table 1). Mouse weight was monitored weekly throughout the experiment, and the changes in body weight were not statistically significant, regardless of CS exposure or BCX supplementation. Urinary cotinine, a metabolite of nicotine, was examined as an indicator of CS exposure. Urinary cotinine levels were significantly higher in CS-exposed groups compared to the non-exposed groups in both WT and DKO mice, regardless of BCX supplementation (*p* < 0.001, Table 1). We observed a small amount of urinary cotinine in mice with no CS exposure. As these levels were lower than in mice with CS exposure, we believe this to be an endogenous metabolite [34] or due to the sham exposure using the same smoking chamber.

### 2.2. Hepatic Vitamin A and BCX Levels in WT and DKO Mice fed Control and BCX Diets with or without CS Exposure

Hepatic BCX was detected in BCX-supplemented groups in WT and DKO mice but was not detectable in non-supplemented groups in WT and DKO mice (Figure 1A). BCX supplementation resulted in a significant accumulation of BCX in the livers of DKO mice (30−43 nmol BCX/g liver), compared with the livers of WT mice (1.7−10.2 nmol BCX/g liver), irrespective of CS exposure (Figure 1A). BCX supplementation did not result in significant differences in hepatic retinol and retinyl palmitate in DKO mice as compared to the non-supplemented DKO mice, irrespective of CS exposure (Figure 1B,C). On the other hand, in WT mice, BCX supplementation significantly increased hepatic retinol compared to the non-supplemented group, irrespective of CS exposure (*p* < 0.05 for both, Figure 1B). Interestingly, CS-exposed WT mice fed BCX exhibited lower hepatic BCX levels as compared to those in non-exposed WT mice fed BCX (*p* < 0.01, Figure 1A). Moreover, CS-exposed DKO mice had decreased hepatic retinol levels when compared to the non-exposed DKO mice, regardless of BCX intake (*p* < 0.05, Figure 1B). Hepatic apo-10′-carotenoids, including apo-10′-carotenal, -carotenol, and -carotenoic acid, were not detectable in any of the groups. The limit of detection for apo-10′-carotenoids in our HPLC system was 10 pmol.

### 2.3. Effects of BCX Supplementation on CS-Induced Inflammatory Cell Infiltration in WT and DKO Mice

CS exposure significantly increased the alveolar infiltration of inflammatory cells in the lung (Figure 2B), as compared to mice without CS exposure (Figure 2A). CS exposure significantly increased both macrophage and neutrophil accumulation in the lung tissues of WT and DKO mice compared to the non-exposed littermates, irrespective of BCX intake (*p* < 0.001 for all, Figure 3A,B). In CS-exposed WT and DKO mice, BCX supplementation significantly decreased levels of macrophage and neutrophil infiltration in the lung tissues, as compared with non-supplemented littermates (*p* < 0.001 for both, Figure 3A,B). As indicated in Figure 3, BCX supplementation had similar protective effects against inflammatory cell infiltration in WT and DKO mice. We chose to investigate the DKO mice at the molecular level because of our interest, specifically in intact BCX. In the CS-exposed DKO mice, BCX supplementation significantly decreased the gene expression of TNF-α and IL-6 induced by CS exposure as compared to non-supplemented DKO mice (*p* < 0.001 and *p* < 0.01, respectively; Figure 3C,D). These results suggest that BCX ameliorates CS-induced lung inflammation independently of BCO1/BCO2. 

### 2.4. Effects of BCX Supplementation on CS-Induced Enlargement of Alveolar Airspace and Bronchiolar Epithelium Hyperplasia in WT and DKO Mice

Histopathological examination showed that CS exposure induced significant higher lung parenchymal destruction compared with mice without CS exposure, leading to the enlargement of alveolar airspace, a common feature of emphysema (Figure 2C), as compared to mice without CS exposure (Figure 2A). CS exposure also increased the bronchial epithelium with hyperplasia in the lung (Figure 2D), in comparison to mice without CS exposure (Figure 2A). Airspace enlargement was quantified based on the assessment of the mean linear intercept (Lm). In CS-exposed WT and DKO mice, BCX supplementation significantly decreased the Lm when compared to non-supplemented littermates (*p* < 0.001 for all, Figure 4A). Similarly, in CS-exposed WT and DKO mice, BCX supplementation significantly lowered the percentage of the bronchial epithelium with hyperplasia compared to non-supplemented littermates (*p* < 0.001 for all, Figure 4B). Matrix metalloproteinases (MMPs) are important mediators of airspace enlargement and alveolar-wall destruction [35]. In CS-exposed DKO mice, BCX supplementation significantly suppressed the mRNA levels of MMP-2 and MMP-9 induced by CS exposure compared to non-supplemented DKO mice (*p* < 0.05 for both, Figure 4C,D). Our results indicate that BCX inhibits CS-induced enlargement of alveolar airspace and bronchial epithelium hyperplasia, independently of BCO1/BCO2.

### 2.5. Effects of BCX and BCX Metabolite, 3OH-BA10C, on Inflammatory Responses in Human Bronchial Epithelial Cells Stimulated with Lipopolysaccharide (LPS)

We further investigated the effects of BCX and its BCO2 cleavage metabolite, 3OH-BA10C, at comparable doses (1 to 4 μM), on lung inflammation in human bronchial epithelial cells (BEAS-2B) in vitro. BEAS-2B cells were pre-treated with BCX at indicated concentrations (0.1% DMSO as control) for two days and subsequently stimulated with 10 ng/mL LPS overnight. Pretreatment with BCX inhibited LPS-induced elevation of IL-6 and TNF-α transcript levels in a dose-dependent manner (1 µM to 4 µM) (Figure 5A). 3OH-BA10C also inhibited the LPS-stimulated expression of IL-6, TNF-α, and MCP-1 at the mRNA level in BEAS-2B cells in a dose-dependent manner (1 to 4 μM) (Figure 6A). The suppressing effects of BCX and 3OH-BA10C on IL-6 expression were associated with increased the expression of sirtuin1 (SIRT1) protein levels (Figure 5B and Figure 6B, respectively) and reduced AKT phosphorylation (Figure 5C and Figure 6C, respectively) in a dose-dependent manner. Only BCX, not 3OH-BA10C, was able to up-regulate protein level of IκB-α (Figure 5D), an inhibitor of NF-κB, in a dose-dependent matter in BEAS-2B cells, without affecting Erk1/2 MAPKs. We did not detect cleavage of BCX in this culture system; it is unlikely that the inhibition of inflammation by BCX was due to vitamin A or apo-10′-carotenals. 

## 3. Discussion

In the present study, we demonstrated that BCX supplementation might have beneficial roles in the prevention of lung injury caused by CS exposure in mice with or without BCO1/BCO2. While the presence or absence of BCO1 and BCO2 did not affect CS-exposure-induced lung lesion severity, BCX supplementation significantly inhibited CS-induced pathological lung injuries, such as inflammatory cell infiltration, hyperplasia of the bronchial epithelium, and enlarged alveolar airspaces in WT and BCO1/BCO2 DKO mice. The beneficial effects of BCX feeding were also associated with suppression of CS-induced gene expression of pro-inflammatory cytokines (TNF-α and IL-6) and MMP-2/9 in the lung tissues of BCO1/BCO2 DKO mice. The present study provided strong experimental supporting evidence and described potential mechanisms to explain the significant protective association between a high intake of BCX and the risk of lung diseases in smokers. Furthermore, we demonstrated that the protective effects of BCX could be a result of the actions of BCX rather than its enzymatic cleavage metabolites of BCO1/BCO2. However, future human intervention studies are needed to identify role of BCO1/BCO2 genotypes in lung cancer development and establish dietary recommendations for carotenoids [36].

Although it has been established that an oxidative burden, such as CS, can cause weight loss in murine models [37], in our study, the final body weight remained unchanged; there was no statistically significant difference among any groups of WT and DKO mice. While it is possible we did not see weight changes due to the short-term nature CS exposure, this observation is important because it suggests that the pathological and molecular changes we investigated are due solely to damaging effects of CS exposure or protective effects of BCX feeding, regardless of BCO1/BCO2 genotypes, respectively. Additionally, a previous report did show sex differences in carotenoid absorption following fruit and vegetable consumption [38]; however, we did not observe any sex difference in terms of BCX accumulation and vitamin A levels in WT or DKO mice. 

It is well known that CS promotes inflammation by increasing immune cells in the airway and inducing the pro-inflammatory cytokines. A clinical trial using transcriptomic analysis demonstrated that many genes that were changed by short-term CS were altered similarly in epithelial cells subject to chronic smoke exposure in the airway in humans [39]. In the current study, 14 days of CS exposure resulted in significant macrophage and neutrophil infiltration in the lungs, and BCX supplementation (including pre-fed BCX for 14 days in order to increase tissue levels of BCX) was able to block this effect in both WT and DKO mice, compared to non-supplemented littermates. These results indicate that BCO1/BCO2 may not play a role in CS-induced lung lesions and that intact BCX possesses anti-inflammatory activities in response to CS exposure. In line with this notion, BCX supplementation significantly decreased CS-exposure-mediated upregulation of TNF-α and IL-6 mRNA expression in the lungs of DKO mice; therefore, BCX-mediated reduction of hyperplasia of the bronchial epithelium may be due to the anti-inflammatory effects of BCX in both WT and DKO mice. This is in agreement with our previous demonstration of BCX’s ability to decrease tobacco-specific carcinogens and nicotine-driven IL-6 mRNA expression and emphysema in the lung [40]. Together, our findings suggest that BCX itself has a unique role in suppressing CS-caused immune cell infiltration and pro-inflammatory cytokines in the lung, which may contribute to the mitigation of lung damage by CS exposure. 

The functional airway epithelium is vital for protection against pathogens. It is well known that CS exposure causes morphological changes to the entire respiratory tract’s epithelial cells, such as hyperplasia and metaplasia. Inhibition of pro-cytokine production by administrating a pro-inflammatory-cytokine antagonist effectively inhibited CS-induced epithelial hyperplasia and metaplasia in mice [41]. In this study, we observed that BCX feeding significantly inhibited CS-induced enlarged alveolar airspaces and hyperplasia of the bronchial epithelium in both WT and DKO mice, although we could not detect any significant metaplasia and emphysema in the lungs of WT and DKO mice due to the short-term nature of the CS exposure. Of note, we demonstrated that BCX supplementation significantly suppressed the CS-induced gene expression of MMP-2/9 in the lungs of DKO mice. MMPs are known to regulate many biological processes, such as immune responses, proliferation, cell death, and emphysema [42]. It has been suggested that macrophages produce elastolytic proteinases such as MMPs, thereby leading to the degradation of alveolar elastin and emphysema [42]. Mice lacking metalloproteinase inhibitors exhibited spontaneous air-space enlargement. Based on this notion, it is plausible that BCX-mediated suppression of macrophage and neutrophil infiltration in the lung may have contributed to the decreased MMP production by immune cells, thereby leading to the amelioration of enlarged alveolar airspaces. This is in agreement with our previous long-term study, which showed that BCX was able to decrease NNK/nicotine-induced emphysema in AJ mice [40] and inhibit the expression of MMP-2 in vitro [14], and retinoic acid’s inability to protect against emphysema development in mice [43]. Our recent study also showed that BCX feeding significantly suppressed high-sugar diet-promoted-hepatic tumorigenesis in both WT and DKO mice, and one of mechanisms was in BCX’s ability to inhibit MMP-2/9 in the hepatic tumors [33]. Altogether, these findings suggest that BCX, without the generation of its biologically active metabolites by BCO1 and BCO2, is responsible for conferring protective properties against lung injury by inhibiting MMP-2/9 in the lung. 

The previous reports in humans showed that people who consumed approximately 0.7 mg/day of BCX had a significantly lower risk of lung cancer [10,11]. We used a physiological dose (20 mg/kg diet) of BCX supplementation, which is equivalent to daily human consumption of approximately 1.7 mg of BCX. These doses of BCX can be easily achieved by daily consumption of 3–4 ounces of a sweet red pepper (BCX 1.3–2.0 mg) or a tangerine (BCX 0.5–0.8 mg). HPLC results showed that BCX feeding resulted in significant hepatic accumulation of BCX (30−43 nmol BCX/g liver) in DKO mice compared to WT mice given BCX supplementation (1.7–10.2 nmol BCX/g liver). The hepatic BCX levels in BCX-fed WT mice were comparable to those in the liver tissues of humans (≈2.3 ± 4.8 nmol BCX/g liver) [44]. We found that BCX feeding resulted in significant changes in hepatic retinol in WT mice, but not in DKO mice, compared to non-supplemented littermates. These results are aligned with our previous study demonstrating that dietary BCX feeding (10 mg/kg diet) resulted in significant accumulation of BCX and yellow coloration of the liver and adipose tissues of DKO mice, but not in WT mice [33], indicating the conversion of BCX to its metabolites by BCO1/BCO2. We only examined BCX levels in the liver because total carotenoid concentrations have been shown to be highest in the liver of all tissues in humans, and to reserve the limited mouse lung tissue samples for other experimental outcomes in this study. In our previous study, we showed that a CS-exposed ferret model, which is more relevant to humans in terms of carotenoid absorption and accumulation [6], fed BCX, had significantly lower levels of BCX in the lung compared to non-exposed ferrets fed BCX [13]. Similarly, in the present study, there were significant decreases in hepatic BCX and retinol in CS-exposed DKO mice compared to non-exposed DKO mice fed BCX. The decreases in BCX and retinol could be explained, in part, due to instability or enhanced degradation of carotenoids and retinoids in a CS-exposed environment [7,22,45]. We also observed a significant decrease in the level of hepatic retinol in CS-exposed groups, regardless of BCX supplementation in DKO mice. Since the AIN-93M semi-purified diet contained sufficient vitamin A, this suggests the preventive effect of BCX on the lung lesions was likely due to the activity of BCX itself rather than the generation of retinol. This is in agreement with our previous observations, which showed that the supplementation of BCX reduced the multiplicity of the carcinogen-induced lung tumor without altering levels of retinol and retinyl ester in both plasma and lung tissue [14].

In this study, however, we were not able to detect any apo-carotenoids, such as 3OH-BA10C, a specific BCO2 cleavage product from BCX. This could have been due to the relative low concentrations or further metabolism of apo-carotenoids, as demonstrated by Kelly et al. [46]. Since the biological function of 3OH-BA10C is largely unknown, we assessed the biological function of 3OH-BA10C, as compared with BCX, using a human bronchial epithelium cell line in vitro. We found that both BCX and 3OH-BA10C treatment prevented LPS-induced inflammatory responses in a dose-dependent manner. The anti-inflammatory effects of BCX and 3OH-BA10C were associated with the upregulation of sirtuin 1 (SIRT1), an NAD^+^-dependent protein/histone deacetylase that has been implicated in various biologic processes, such as metabolism, inflammation, and immune function in lung disease [47]. We have previously revealed that BCX supplementation restored SIRT1 along with lung inflammation and tumorigenesis in A/J mice [40]. However, because comparable doses of BCX and 3OH-BA10C can attenuate LPS-induced inflammatory responses, the modulation of SIRT1 with BCX against inflammation could be due to the biological action of BCX as its own intact molecule. This was supported in that only BCX, not 3OH-BA10C, was able to up-regulate the protein level of IκB-α, a negative regulator of NF-κB. Although we provided the first in vitro evidence that 3OH-BA10C has an anti-inflammatory function in human bronchial epithelium, potentially by inducing SIRT1, it is unlikely that the inhibition of the LPS-induced inflammatory responses by BCX was due to its conversion into vitamin A or apo-10-carotenoids due to the low concentrations of BCX (1 to 4 µM) used in this in vitro study, which supports the notion that BCX has its own biological function, as shown in our in vivo study. A more thorough examination of how BCX and 3OH-BA10C may differentially affect IκB-α, which diminishes inflammation, is required and warrants greater attention.

## 4. Materials and Methods

### 4.1. Animals, Diets, and CS-Exposure

Male and female BCO1/BCO2 DKO mice with a C57BL/6J genetic background and wild type (WT) mice were randomly divided into 4 groups: (1) control; (2) control + 20 mg BCX/kg diet; (3) CS exposure; (4) CS exposure + 20 mg BCX/kg diet. At 25 weeks of age, mice were fed BCX (>99% purity, BASF, Ludwigshafen, Germany) directly mixed with an AIN-93M semi-purified diet (20 mg BCX/kg diet), as previously described [14], for 2 weeks. At 27 weeks of age, mice received whole body, mainstream CS exposure (Research Cigarettes, Type 3R4F, at a rate of 10 cigarettes/30 min, twice in the morning and twice in the afternoon) for 14 days, as described [13,48,49,50]. This amount of tobacco smoking-exposure in the animals is similar to that found in humans smoking one and a half packages of cigarettes per day, as measured by the concentration of urinary cotinine equivalents, which was confirmed in this study. Urinary cotinine was measured by using an ELISA kit (ABNOVA, Taipei, Taiwan) as a CS-exposure index [13]. The control groups without CS exposure underwent sham exposure using the same smoking chamber. The supplementation of BCX at 20 mg/kg diet was given daily to the mice starting two-weeks prior to the CS exposure and continued for 2 weeks during the CS exposure. The body weights of the mice were recorded weekly. This study was approved by the Animal Care and Use Committee at the HNRCA at Tufts University.

### 4.2. Histological Analysis

The right upper lobe of each lung was inflated and fixed by intratracheal infusion of 10% neutral buffered formalin at 15 cm of H_2_O pressure for 2 min and then immersed in fresh 10% neutral buffered formalin for 72 h, as previously described [14]. The samples were subsequently processed and embedded in paraffin, and serially sectioned. Four-micrometer sections were stained with hematoxylin (H) and eosin (E) for histopathological examinations. Hyperplasia of bronchial epithelium was examined for 20 fields at a magnification of ×100 (0.63 cm^2^) for each animal and expressed as percent of bronchi with hyperplastic epithelium (the number of bronchi with hyperplastic epithelium divided by the total number of bronchiolar examined). To evaluate lung inflammation, macrophages and neutrophils were quantified in the alveoli for 10 fields at a magnification of ×400 (1.59 mm^2^) for each animal, and the number of macrophages or neutrophils was expressed as cells/mm^2^. The quantification of airspace enlargement was determined by the mean linear intercept (Lm) in micrometers. The measurement of Lm was performed by using a 100 × 100 μm grid that was randomly positioned over each field in the lung. The total length of each line of the grid divided by the number of alveolar intercepts yielded the average distance between alveolates surfaces, or the Lm. For each animal, 10 fields at a magnification of ×200 were examined. The sections were examined blindly by independent investigators by light microscopy (ZEISS Microscopy, Dublin, California, USA), equipped with PixeLINK USB 2.0 (PL-B623CU) camera, and PixeLINK μScope Microscopy Software, Version 1.0).

### 4.3. High-Performance Liquid Chromatography (HPLC)

Liver samples of mice were prepared as previously described [14,40]. Samples were reconstituted with 100 μL of a 2:1 ethanol:methyl-tert butyl ether solution. A gradient reverse-phase HPLC system consisting of a Waters 2695 separation module and a Waters 2998 photodiode array detector was used for the detection of BCX, retinol, retinyl palmitate, and retinoic acid. BCX, retinol, retinyl palmitate, and retinoic acid were analyzed on a reverse-phase C18 column (4.6 × 250 mm, 5 μm) (Vydac 201TP54, Grace Discovery Sciences, Inc., Bannockburn, IL, USA) with a flow rate of 1.00 mL/min and quantified relative to internal standards by determining the peak areas against known amounts of standards.

### 4.4. Cell Culture and Reagents

All-trans BCX (>99% purity, BASF, Ludwigshafen, Germany) and 3-OH-β-apo-10′-carotenal (3OH-BA10C, >99% purity, BASF, Ludwigshafen, Germany) were applied to human bronchial epithelial cells, BEAS-2B (ATCC, Manassas, VA, USA), on plates pre-coated with bovine serum albumen/collagen/fibronectin with serum-free LHC-9 medium. Cells were pre-treated with BCX or 3OH-BA10C for 24 h and were stimulated with 10 ng/mL lipopolysaccharide (LPS), an established endotoxin, overnight, and the cell pellets were collected.

### 4.5. Quantitative Real-Time Polymerase Chain Reaction (qRT-PCR)

Total RNA was extracted using TriPure Isolation Reagent (Roche Applied Science). cDNA was prepared using M-MLV reverse transcriptase (Invitrogen, Carlsbad, CA, USA) and an automated thermal cycler, PTC-200 (MJ Research, Bio-Rad Laboratories, Hercules, CA, USA). qRT-PCR was carried out using a Fast Start Universal SYBR Green Master (ROX) (Roche, Indianapolis, IN, USA). Relative gene expression was determined using the 2^−ΔΔCt^ method.

### 4.6. Western Blotting

Sample proteins were resolved by SDS-PAGE and transferred onto Immobilon-P membranes (Millipore Corp., Bedford, MA, USA). The primary antibodies phosphorylated-Akt (Ser473), total Akt, total IkBa (Cell Signaling), and Sirt1 (Santa Cruz), were used. Protein was detected by horseradish peroxidase-conjugated secondary antibody (Bio-Rad, Hercules, CA, USA) and visualized by a SuperSignal West Pico Chemiluminescent Substrate Kit (Pierce, Rockford, IL, USA).

### 4.7. Statistical Analysis

Measurements are expressed as the mean ± the SEM, unless otherwise indicated. Comparisons across multiple groups were conducted by one-way ANOVA with Tukey’s Honestly Significant Difference (HSD). Differences between two groups were analyzed by the Student’s t-test. Analyses were performed using GraphPad Prism (Version 7.01).

## Figures and Tables

**Figure 1 molecules-28-01383-f001:**
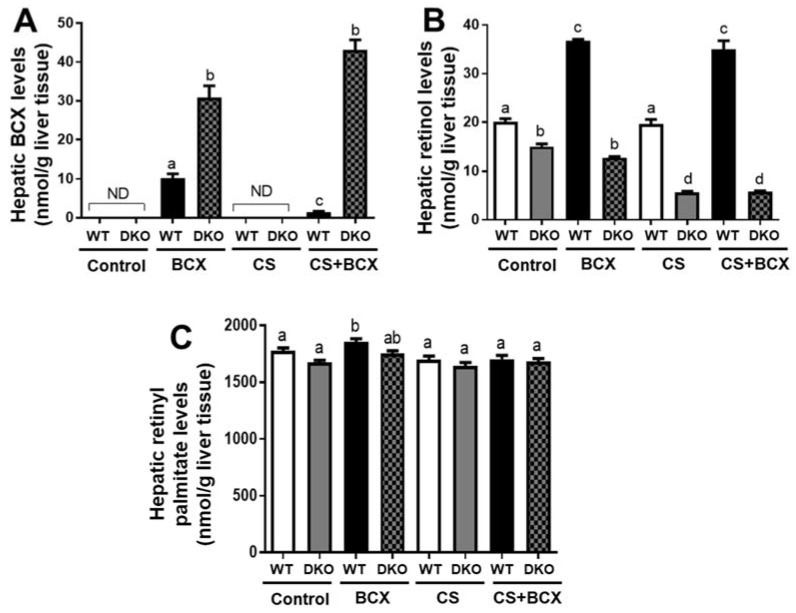
Hepatic BCX, retinol, and retinyl palmitate in WT and *BCO1^−/−^/BCO2^−/−^* DKO mice. Hepatic levels of (**A**) BCX, (**B**) retinol, and (**C**) retinyl palmitate in mice, detected by HPLC. Groups: Control, non-CS-exposed mice without supplementation; BCX, non-CS-exposed mice with BCX supplementation; CS, CS-exposed mice without supplementation; CS+BCX, CS-exposed mice with BCX supplementation. Significant differences between groups were analyzed by one-way ANOVA and are denoted by different letters (significance: *p* < 0.05). BCX, β-cryptoxanthin; CS, cigarette smoke; HPLC, high-performance liquid chromatography; WT, wild type; DKO, double knockout; BCO1, β-carotene-15,15′-oxygenase; BCO2, β-carotene-9′,10′-oxygenase; ND, Not detected.

**Figure 2 molecules-28-01383-f002:**
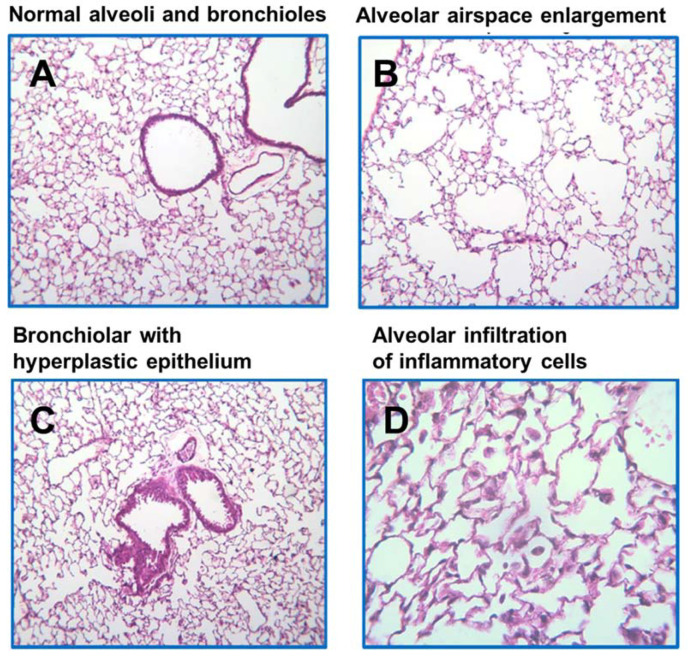
Histopathologic examination of lung tissues of *BCO1^−/−^/BCO2^−/−^* DKO mice exposed to CS or not. Representative image of an H&E-stained lung at 40× (four-micrometer sections). (**A**) normal alveoli and bronchiolar of non-CS-exposed control group. Fourteen days of CS exposure significantly induced (**B**) alveolar airspace enlargement, (**C**) bronchiolar with hyperplastic epithelium, and (**D**) alveolar infiltration of inflammatory cells in DKO mice. H&E, hematoxylin and eosin. CS, cigarette smoke; BCO1, β-carotene-15,15′-oxygenase; BCO2, β-carotene-9′,10′-oxygenase; DKO, double knockout.

**Figure 3 molecules-28-01383-f003:**
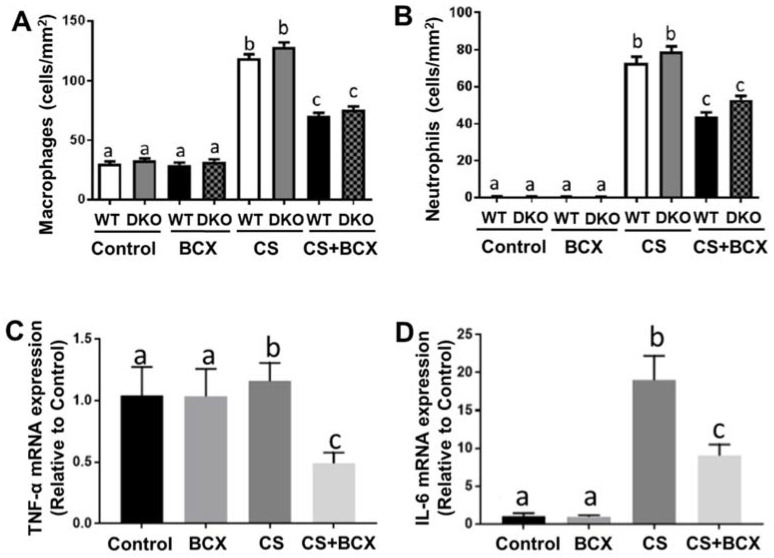
BCX feeding significantly prevented inflammation in the lung of WT mice and *BCO1^−/−^/BCO2^−/−^* DKO mice exposed to CS. (**A**) Quantification of H&E stained (at 10×) lung pathology of macrophage accumulation, and (**B**) neutrophil accumulation in the lungs of WT and DKO mice. (**C**) Shown are the graphical representations of mRNA expression of C) TNF-α and (**D**) IL-6 quantified by real-time PCR in the lungs of DKO mice (*n* = 7–9 for each group). β-actin was used as a loading control. Groups: Control, non-CS-exposed mice without supplementation; BCX, non-CS-exposed mice with BCX supplementation; CS, CS-exposed mice without supplementation; CS+BCX, CS-exposed mice with BCX supplementation. Lung inflammation, macrophages, and neutrophils were quantified in the alveoli for 10 fields at a magnification of 400× (1.59 mm^2^) for each animal; and the number of macrophages or neutrophils is expressed as cells/mm^2^. Significant differences between groups were analyzed by one-way ANOVA and are denoted by different letters (significance: *p* < 0.05). BCX, β-cryptoxanthin; CS, cigarette smoke; WT, wild type; DKO, double knockout; BCO1, β-carotene-15,15′-oxygenase; and BCO2, β-carotene-9′,10′-oxygenase.

**Figure 4 molecules-28-01383-f004:**
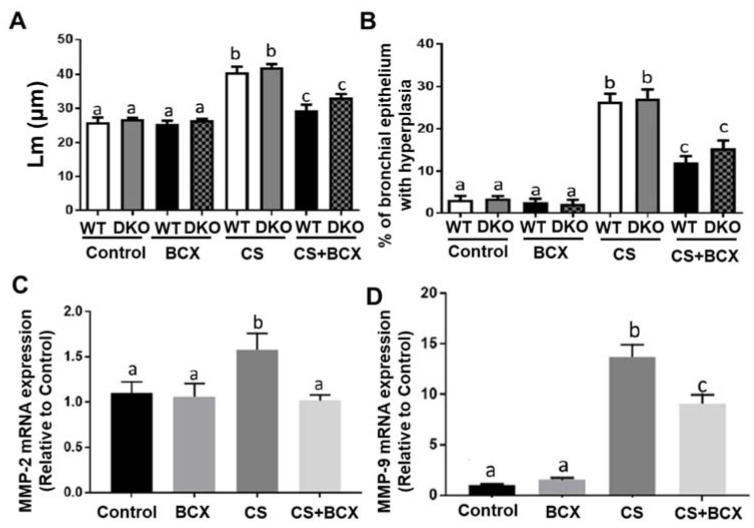
BCX supplementation significantly inhibited CS-exposure-induced emphysema and proliferative lung lesions in WT mice and *BCO1^−/−^/BCO2^−/−^* DKO mice. Quantification of H&E stained (at 10×) lung pathology of (**A**) alveolar diameter expressed as mean linear intercept (Lm, μM) and (**B**) percentage of hyperplastic of the bronchial epithelium in WT and DKO mice. Shown are the graphical representations of mRNA expression of (**C**) MMP-2 and (**D**) MMP-9 quantified by real-time PCR in the lungs of DKO mice (*n* = 7–9 for each group). β-actin was used as a loading control. Groups: Control, non-CS-exposed mice without supplementation; BCX, non-CS-exposed mice with BCX supplementation; CS, CS-exposed mice without supplementation; CS+BCX, CS-exposed mice with BCX supplementation. Significant differences between groups were analyzed by one-way ANOVA and are denoted by different letters (*p* < 0.05). BCX, β-cryptoxanthin; CS, cigarette smoke; DKO, double knockout; BCO1, β-carotene-15,15′-oxygenase; BCO2, β-carotene-9′,10′-oxygenase; and MMP, matrix metalloproteinase.

**Figure 5 molecules-28-01383-f005:**
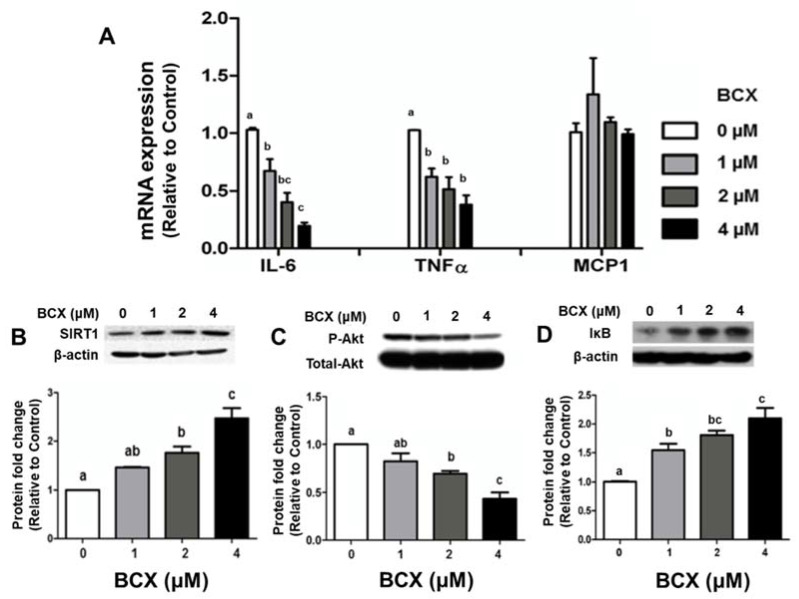
BCX treatment reduced inflammation, increased SIRT1 protein, and decreased phosphorylation of Akt in LPS-stimulated BEAS-2B cells. Shown are the graphical representations of mRNA expression of (**A**) IL-6, TNF-α, and MCP-1 quantified by real-time PCR. β-actin was used as a loading control. Protein expression levels of (**B**) SIRT1, (**C**) p-Akt, and (**D**) IκBα were measured by Western blot in BEAS-2B cells pre-treated with BCX (0.5 to 4 μM) following LPS stimulation (10 ng/mL). β-actin and total-Akt were used as controls. Significant differences between groups were analyzed by one-way ANOVA and are denoted by different letters (*p* < 0.05). BCX, β-cryptoxanthin; SIRT1, sirtuin 1; BEAS-2B, human bronchial epithelial cells; TNF, tumor necrosis factor; IL, interleukin; MCP-1, monocyte chemoattractant protein 1; SIRT1, sirtuin1; AKT, protein kinase B; IκBα, nuclear factor of kappa light polypeptide gene enhancer in B-cells inhibitor, alpha; LPS, lipopolysaccharide.

**Figure 6 molecules-28-01383-f006:**
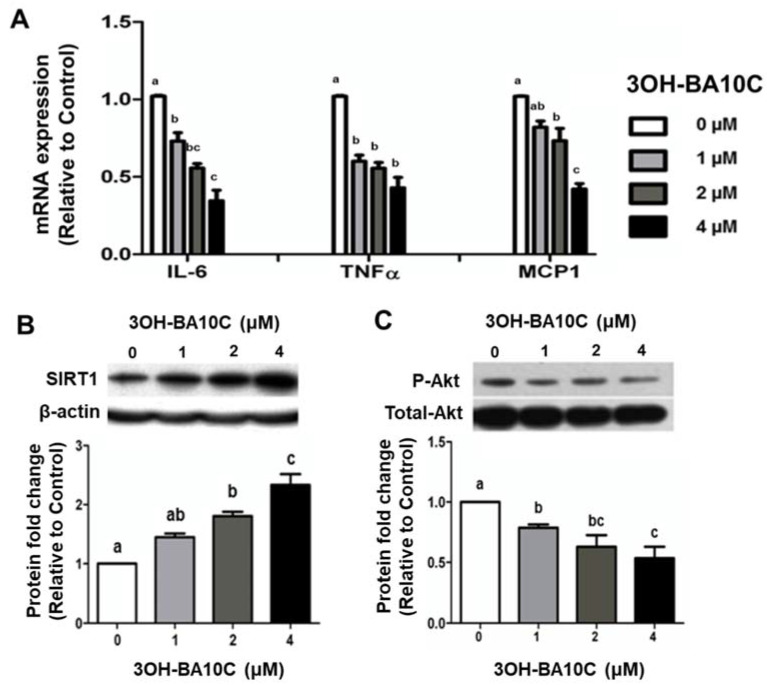
**The** BCX metabolite 3-OH-β-apo-10’-carotenal (3OH-BA10C) protected against inflammation, increased SIRT1 protein, and decreased phosphorylation of Akt in LPS-stimulated BEAS-2B cells. Shown are the graphical representations of mRNA expression of (**A**) IL-6, TNF-α, and MCP-1 quantified by real-time PCR. β-actin was used as a loading control. Protein expression levels of (**B**) SIRT1 and (**C**) p-Akt were measured by Western blot in BEAS-2B cells pre-treated with 3OH-BA10C (0.5 to 4 μM) following LPS stimulation (10 ng/mL). β-actin and total-Akt were used as controls. Significant differences between groups were analyzed by one-way ANOVA and are denoted by different letters (*p* < 0.05). BCX, β-cryptoxanthin; SIRT1, sirtuin 1; TNF-α, tumor necrosis factor-α; IL, interleukin; MCP-1, monocyte chemoattractant protein 1; SIRT1, sirtuin 1; AKT, protein kinase B; LPS, lipopolysaccharide.

**Table 1 molecules-28-01383-t001:** Body weight (BW) and urinary cotinine (UC) of cigarette smoke (CS)-exposed wild type and BCO1/BCO2 double knockout (DKO) mice with or without β-cryptoxanthin (BCX) treatment.

	Wild Type				BCO1/BCO2 DKO			
Experimental Group	Control	BCX	CS	CS+BCX	Control	BCX	CS	CS + BCX
Animal (*n*)	6	6	6	6	8	7	9	8
Male	3	3	3	3	5	4	5	4
Female	3	3	3	3	3	3	4	4
Age (day)	206	206	206	206	200	201	205	210
Final BW (g)	36 ± 9	37 ± 7	38 ± 8	35 ± 7	33 ± 9	34 ± 6	31 ± 6	32 ± 8
UC (μg/mL)	0.5 ± 0.2 ^a^	0.4 ± 0.2 ^a^	3.9 ± 1.3 ^b^	4.0 ± 3.1 ^b^	0.4 ± 0.4 ^a^	0.2 ± 0.2 ^a^	4.1 ± 2.3 ^b^	4.0 ± 3.2 ^b^

Data are represented as the mean ± SEM. Significant differences between groups were analyzed by one-way ANOVA, and different letters denote significance at *p* < 0.05.

## Data Availability

All data including digital formats are maintained in the archive folder on our lab’s shared network drive and are available from the corresponding author on request from the scientific research community.

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
