# Peer review of "β-Cryptoxanthin Attenuates Cigarette-Smoke-Induced Lung Lesions in the Absence of Carotenoid Cleavage Enzymes (BCO1/BCO2) in Mice"

_molecules, 2023, doi:10.3390/molecules28031383_

Round 1
Reviewer 1 Report
1. how the animals were treated with the test samples.
2. what is the rartionale of this study.
3. the important findings must be updated.
Author Response
Reviewer 1:
- How the animals were treated with the test samples.
Response: Thank you. We have added “The supplementation of BCX at 20 mg/kg diet was given daily to the mice starting two-weeks prior to the CS-exposure and continued for 2 weeks during the CS-exposure”. Please see the method section, line 396-398, in the revised manuscript.
- What is the rationale of this study.
Response: Thank you. We have outlined more clearly the rationale of this study (see line 84-87 and line 94-99). Specifically, we aimed to understand whether carotenoid cleavage enzymes (BCO1/BCO2 genotype) modify the effects of BCX supplementation in smokers, and to further understand whether the effect of BCX is a result of the actions of BCX or its enzymatic cleavage metabolites (see line 100 to 104)
- The important findings must be updated.
Response: We have indicated the important findings in both the graphic abstract and in the manuscript (line 261-271). In the revised manuscript, we have highlighted further that the effect of BCX could be a result of the actions of BCX rather than its enzymatic cleavage metabolites of BCO1/BCO2 (see line 272).

Reviewer 2 Report
This is a very carefully prepared and described in vivo and in vitro study of the impact of BCX on cigarette smoke-induced lung lesions. It follows up on the prior work from this group. There are only a few suggestion/questions.
1. Table 1. It is not clear why there are two control groups within WT mice listed in the table. Please clarify.
2. Lines 135 - 136. The authors should provide the limit of detection for apo-10'-carotenoids
3. Line 195. Spelling typo "lowered"
4. Lines 358 - 360. Please reword sentence - missing a word.
Author Response
Reviewer 2:
This is a very carefully prepared and described in vivo and in vitro study of the impact of BCX on cigarette smoke-induced lung lesions. It follows up on the prior work from this group. There are only a few suggestion/questions.
- Table 1. It is not clear why there are two control groups within WT mice listed in the table. Please clarify.
Response: Thank you for your thorough review. We believe that in Table 1 of the original version the line under wild type was somehow extended to the control group in BCO1/BCO2 DKO mice. . We had only one control group in WT and DKO, respectively. We corrected this in the revised Table 1 and apologize for the confusion
- Lines 135 - 136. The authors should provide the limit of detection for apo-10'-carotenoids
Response: Agreed. The limit of detection for apo-10'-carotenoids in our HPLC system was 10 pmol. We have added the information in line 140
- Line 195. Spelling typo "lowed" Response: agreed and corrected.
- Lines 358 - 360. Please reword sentence - missing a word. Response: agreed and corrected.
